# Features of Helium and Tritium Release from Li_2_TiO_3_ Ceramic Pebbles under Neutron Irradiation

**DOI:** 10.3390/ma16175903

**Published:** 2023-08-29

**Authors:** Timur Kulsartov, Zhanna Zaurbekova, Yevgen Chikhray, Inesh Kenzhina, Saulet Askerbekov, Asset Shaimerdenov, Assyl Akhanov, Magzhan Aitkulov, Meiram Begentayev

**Affiliations:** 1Satbayev University, 22 Satbayev str., Almaty 050013, Kazakhstan; tima@physics.kz (T.K.); kenzhina@physics.kz (I.K.); m.begentayev@satbayev.university (M.B.); 2Institute of Applied Sciences and Information Technologies, 280 Bayzakov str., Almaty 050040, Kazakhstan; chikhray@physics.kz (Y.C.); askerbekov@physics.kz (S.A.); 3Institute of Experimental and Theoretical Physics, Al-Farabi Kazakh National University, 71/23 Al-Farabi ave., Almaty 050040, Kazakhstan; 4Al-Farabi Kazakh National University, 71/23 Al-Farabi ave., Almaty 050040, Kazakhstan; 5Institute of Nuclear Physics, 1 Ibragimov str., Almaty 050032, Kazakhstan; ashaimerdenov@inp.kz (A.S.); aakhanov@inp.kz (A.A.); maitkulov@inp.kz (M.A.); 6Kazakh-British Technical University, 59 Tole bi, Almaty 050000, Kazakhstan

**Keywords:** lithium ceramics, helium, tritium, neutron irradiation, desorption

## Abstract

The operation of fusion reactors is based on the reaction that occurs when two heavy hydrogen isotopes, deuterium and tritium, combine to form helium and a neutron with an energy of 14.1 MeV D + T → He + n. For this reaction to occur, it is necessary to produce tritium in the facility itself, as tritium is not common in nature. The generation of tritium in the facility is a key function of the breeder blanket. During the operation of a D–T fusion reactor, high-energy tritium is generated as a result of the ^6^Li(n,α)T reaction in a lithium-containing ceramic material in the breeder blanket. Lithium metatitanate Li_2_TiO_3_ is proposed as one of the promising materials for use in the solid breeder blanket of the DEMO reactor. Several concepts for test blanket modules based on lithium ceramics are being developed for testing at the ITER reactor. Lithium metatitanate Li_2_TiO_3_ has good tritium release parameters, as well as good thermal and thermomechanical characteristics. The most important property of lithium ceramics Li_2_TiO_3_ is its ability to withstand exposure to long-term high-energy radiation at high temperatures and across large temperature gradients. Its inherent thermal stability and chemical inertness are significant advantages in terms of safety concerns. This study was a continuation of research regarding tritium and helium release from lithium metatitanate Li_2_TiO_3_ with 96% ^6^Li during irradiation at the WWR-K research reactor using the vacuum extraction method. As a result of the analysis of experiments regarding the irradiation of lithium metatitanate in vacuum conditions, it has been established that, during irradiation, peak releases of helium from closed pores of the ceramics are observed, which open during the first 7 days of irradiation. The authors assumed that the reasons samples crack are temperature gradients over the ceramic sample, resulting from the internal heating of pebbles under the conditions of their vacuum evacuation, and contact with the bottom of the evacuated capsule. The temperature dependence of the effective diffusion coefficient of tritium in ceramics at the end of irradiation and the parameters of helium effusion were also determined.

## 1. Introduction

The operation of fusion reactors is based on the reaction that occurs when two heavy hydrogen isotopes, deuterium and tritium, combine to form helium and a neutron with an energy of 14.1 MeV:D + T → He + n.(1)

For this reaction to occur, it is necessary to produce tritium in the facility itself, as tritium is not common in nature. The generation of tritium in the facility is a key function of the breeder blanket. During the operation of a D–T fusion reactor, high-energy tritium is generated as a result of the ^6^Li(n,α)T reaction in a lithium-containing ceramic material in the breeder blanket. In the DEMO reactor, the main task of which is to demonstrate the feasibility of power plants based on a fusion reactor, the breeder blanket also removes thermal energy from the first wall and provides neutron shielding.

Lithium metatitanate Li_2_TiO_3_ is proposed as one of the promising materials for use in the solid breeder blanket of the DEMO reactor. Several concepts of test blanket modules based on lithium ceramics are being developed for testing at the ITER reactor [1]. Lithium metatitanate Li_2_TiO_3_ has good tritium release parameters, as well as good thermal and thermomechanical characteristics [2,3]. The most important property of lithium ceramics Li_2_TiO_3_ is its ability to withstand exposure to long-term high-energy radiation at high temperatures and across large temperature gradients. Its inherent thermal stability and chemical inertness are significant advantages in terms of safety concerns. The properties exhibited by candidate materials are the subject of extensive research, as they are critical to the realization of numerous conceptual designs for solid ceramic breeders.

This study was a continuation of research [4,5,6,7] regarding tritium and helium release from lithium metatitanate Li_2_TiO_3_ with 96% ^6^Li during irradiation at the WWR-K research reactor using the vacuum extraction method. Thus, in [4], the initial section of the experiment (the stepwise increase in reactor power) was analyzed; it was shown that the release of tritium occurred in the form of HT and T_2_ molecules. In [5], the nature of tritium-containing molecules release was analyzed over the entire period of irradiation; it was found that tritium was released uniformly, except for areas where the irradiation conditions changed noticeably. In [6,7], the parameters of the Arrhenius dependence of the tritium effective diffusion coefficient were estimated during a sample’s heating at a stationary reactor power and during short-term (5–10 min) reactor power reductions.

In this paper, the analysis of tritium and helium release from lithium metatitanate Li_2_TiO_3_ during irradiation is considered, taking into account new data regarding the nature of M4 (HT and He) mass release obtained later in reactor experiments using two-phase lithium ceramics Li_4_SiO_4_-Li_2_TiO_3_ [8].

## 2. Materials and Methods

A total of 177 lithium ceramics Li_2_TiO_3_ pebbles, with an average diameter of ~1 ± 0.5 mm and a total weight of 0.37 g (with a sphericity of ~1.05, a density of ~1.37 g/cm^3^ and a porosity of ~7%), were considered. Lithium enrichment with ^6^Li isotope was 96%. Ceramic pebbles were obtained via sol–gel technology. The elemental composition of the material is presented in Table 1.

The loading capsule with test samples (Figure 1) was placed in a pumped-out experimental ampoule device connected to the CIRRA (Complex of In-Reactor Release Analysis) experimental facility (Figure 2) located at the WWR-K research reactor in Almaty, Kazakhstan [9]. Ceramic pebbles were poured into the bottom of the experimental ampoule device in 1 layer. The calculations performed showed that the self-shielding effect in the pebble bed was less than 10%. The samples were located at the center level of the reactor core. The thermal neutron flux density in the irradiation point was approximately 5 × 10^13^ n/(cm^2^·s).

The vacuum extraction experiment consisted of the following: samples of lithium ceramics Li_2_TiO_3_ were irradiated under continuous pumping out while the release of tritium-containing molecules (HT, T_2_, HTO and T_2_O) was recorded using a mass spectrometry system. The average temperature of the samples during irradiation at 6 MW of reactor power was ~610 °C; the irradiation procedure continued for 21 days.

It is important to note that during irradiation experiments at the WWR-K reactor, a partial change in the irradiation channels was carried out (lasting less than 10 min once or twice a day). Therefore, the reactor power level was reduced by 25–50% during this period, and then the power was restored to its previous value. Correspondingly, the sample temperature and the fluxes of released tritium-containing gases decreased for a short time.

## 3. Results and Discussion

### 3.1. Results of the Reactor Experiment

The results of the presented reactor experiment have been partially considered in [4,5,6,7]. In [4], the initial section of the experiment (the stepwise output section of the reactor) was analyzed; it was shown that the tritium release occurred in the form of HT and T_2_ molecules. The ratio of tritium-containing gas fluxes depended on the hydrogen pressure during the gas phase (section I in Figure 3).

Further, in [5], the release pattern of tritium-containing molecules over the entire irradiation period was analyzed; it was found that tritium was released uniformly, except in sections where the irradiation conditions changed noticeably. When returning to normal irradiation conditions, the fluxes of released gases also returned to some trend values. It was also confirmed that the main amount of tritium during the whole irradiation was released in the form of HT molecules. However, the flux of the HT molecule decreased with time, due to the fact that the release of the T_2_ molecule flux increased.

Sections of the experiment associated with changes in irradiation conditions, namely, heating samples at steady-state reactor power and short-term (5–10 min) drops in reactor power, were analyzed in [6,7]. In these works, the parameters of the Arrhenius dependence of the tritium effective diffusion coefficient were estimated (sections II and III in Figure 3).

However, further analysis of the results of the irradiation experiment, presented in this paper, was due to new information regarding the nature of the release of gases with a mass number of M4, which the authors were able to obtain in later reactor experiments conducted using two-phase lithium ceramics [8].

A detailed explanation follows. Sharp decreases in the pressure of M4 gases observed in the graph were due to a decrease in the sample temperature as a result of short-term reductions in reactor power (section III in Figure 3). During the irradiation period, several tens of such short-term power reductions were carried out; accordingly, they were all determined by a decrease in sample temperature, and were separately analyzed in [6]. An enlarged characteristic section III is shown in Figure 4 (it should be mentioned that there were several tens of such sections over the course of the experiment).

Further, as can be seen from Figure 3, the pressure changes in gases with the mass number M4 in the course of the experiments were characterized by frequent peaks relative to a certain trend level. In previous works [4,5,6,7], it was assumed that such a peak release of gases with the mass number M4 was due to methodological peculiarities of measurements, and the peak releases themselves were not analyzed (only the lower trend of release was). However, later, in [8,10], it was shown that this release is associated with helium peak releases from ceramics. In these studies, two mass spectrometers were used at selected time intervals in the experiment in order to exclude methodologically determined errors in the measurements.

When interpreting the obtained results of gas composition changes in the chamber during irradiation, it was suggested that the release of the M4 peak could be decomposed into two components. The first component, which is responsible for the release of the HT molecule, is the lower level of the M4 release curve (trend level in Figure 3). The second component, which is responsible for the variable component, refers to the abrupt release of helium from the inner pores of the ceramic (when the free escape pathways to the pebble surface are formed). The pulse amplitude (peak height) was determined by both the volume of the internal cavity, in which the gas accumulated, and by its limiting pressure, which provided the breakthrough. Here it is also worth noting that the peak releases were characteristic only for the gas with mass M4, and were not observed for other masses.

### 3.2. Analysis of Helium Release from a Ceramic Pebble

Data obtained in experiments using two-phase lithium ceramics [8,10] allowed a new evaluation of the results of the presented experiment using Li_2_TiO_3_ + 5 mol% TiO_2_ ceramics.

First, we plotted the dependence of the appearance rate of helium release peaks of different amplitudes on the irradiation time (Figure 5). Here, the average pressure level for M4 gases was chosen as the level characteristic determining the magnitude of the peak. Further calculation took into account peak releases exceeding this level by at least 20%. The peaks were calculated for the time interval from 0.5 to 2 days using measurement intervals of 6 h, and for time intervals from 2 to 7 days using measurement intervals of 12 h. Data shown in Figure 5 formalize the following features of helium release peaks during the experiment:(1)The greatest number of observed helium release peaks ranged from 20% to 60% of the trend release level; their release rate increased with the irradiation time up to 2–3 days of irradiation, and then decreased;(2)Peaks with a release of more than 80% of the trend release level were mostly observed at the beginning of the experiment;(3)After the 7th day of irradiation, the release of helium in the form of peaks practically stopped.

The mechanism associated with such a release, in our opinion, is as follows. The helium produced in the volume of ceramics does not practically diffuse and cannot be released from the ceramics. However, it can enter an internal pore of the ceramics during the reaction of ^6^Li with neutrons in regions close to these pores. Next, due to kinetic energy, helium atoms enter a pore and accumulate in it during irradiation. When a pore is opened (for example, when the sample cracks slightly), the helium leaves the sample “by a free escape path”, and the value of the release peak is determined by the amount of helium accumulated in the pore, which should be higher by the end of irradiation. Hence, the total amount of helium released as a peak should be proportional to the surface of the inner pore and the irradiation time. Estimating the total amount of helium released in the experiment, the authors estimated the average size of the “inner pores” to be approximately 1–2 µm.

Based on data obtained, it can be formally assumed that the result of such a mechanism is the following pattern of pore opening: at the initial stage of irradiation, larger pores are opened (with a characteristic size up to 10 μm, and more); then, for some time (up to 3–4 days of irradiation), pores with a characteristic size up to 2 μm are mainly opened; next, smaller pores (up to 1 μm) are opened; after that, on days 7–8, the processes of “free paths” formation virtually cease. However, real cracking should appear first when stresses are concentrated on the strongly curved surfaces of closed pores, and also when there is a high helium pressure within the pores.

As for the reasons for sample cracking, the authors assume that cracks are caused mainly by temperature gradients along the ceramic sample under irradiation conditions. The cracking of lithium metatitanate can begin at temperature gradients exceeding critical values, which depend on various factors, such as material composition, sample size and shape, heating and cooling rates, and experimental conditions.

Some studies have shown that temperature gradients can occur in the ceramic pebble bed, especially in the area of ceramic–steel contact, because in the interfacial region additional heat transfer resistance is introduced [11,12]. Neutrons interacting with ceramic oxides can cause induced activation, formation of gas bubbles, reduction in thermal conductivity, embrittlement and other material-degrading effects [13,14]. We suppose these factors can lead to the cracking of lithium metatitanate pebbles located in the contact area.

Unfortunately, we have not yet been able to confirm the proposed helium release mechanism from data of microstructural studies of the pebble cross section (for example, as it was confirmed in [15]) due to the high activity of test samples.

Thermophysical calculations conducted for the lithium metatitanate pebble for irradiation conditions, taking into account the non-uniform distribution of energy release along the pebble radius (due to self-shielding), as well as taking into account vacuum conditions, showed that the observed gradients along the pebble were quite significant; in the area of contact of the sample with the stainless steel capsule bottom they could be up to 80 K/mm (Figure 6).

Next, the section of the experiment related to the moment of reactor stoppage was analyzed (section IV, Figure 3). As can be seen in Figure 7, a large peak of helium release was observed after the beginning of sample cooling. The same peaks were observed by the authors of this paper in four other reactor experiments with two-phase lithium ceramics. This peak was due to the release of helium and not the HT molecule, as the peak release of HT should have been accompanied by a peak release of the T_2_ molecule. Here, it can also be noted that this release of helium was most likely due to the formation of significant temperature gradients across the pebble as a result of a change in its heating mode (when the reactor stopped heating the sample due to the nuclear reaction of the neutron with the ^6^Li atom, the sample stopped being heated).

The release of HT and T_2_ gas fluxes into the system can be represented as follows (according to Polanyi–Wigner’s law [16]):(2)$$\varphi \left(HT\right)\sim b_{1}\cdot C_{H}^{1}\cdot C_{T}^{1},$$
(3)$$\varphi \left(T_{2}\right)\sim b_{2}\cdot C_{T}^{2},$$
where $C_{H}^{1}$ and $C_{T}^{1}$ are concentrations of H and T atoms on the ceramic surface, respectively; $C_{H}^{1}\sim \sqrt{P_{H_{2}}}$); and *b*_1_, *b*_2_ and *b*_3_ are constants of molecule release from the surface-to-gas phase.

From these expressions, we can separate the release of gases with a mass number of M4 into helium and the HT molecule. Figure 8 represents the release of various gases into the experimental chamber, including the total flux of tritium (2T_2_ + HT).

### 3.3. Simulation of Tritium Release and Estimation of the Effusion Parameters of Helium

Next, the obtained dependence of tritium release was simulated within the diffusion mechanism. Helium release was analyzed within the effusion model; i.e., helium exits the cavity with gas through the hole into the vacuum.

A simulation of tritium release was conducted via the finite element method using the COMSOL Multiphysics software environment, assuming that the release of tritium is determined only by diffusion processes. To calculate the equilibrium distribution of tritium in the samples at the time of a reactor stoppage, a time interval of 100,000 s was chosen as the final stage of irradiation. In the calculations, experimental data for reactor power, the tritium production rate, and the sample temperature were entered as input data. The results of the simulation are shown in Figure 9.

The temperature dependence of the effective diffusion coefficient of tritium in ceramics at the end of irradiation was determined based on the simulation results:(4)$$D=1\times 10^{-7}\left(\frac{m^{2}}{s}\right)\exp\left(-\frac{120\left(\frac{kJ}{mole}\right)}{RT}\right).$$

The dependence of helium release was used to estimate the effusion parameters; in particular, the probability values (*β*) of the helium atom to leave the internal pores through “free paths” formed immediately after the reactor shutdown.

The change in helium atoms quantity in the inner pores volume can be represented as:(5)$$N(t)=N(t_{0})\cdot e^{-\beta t},$$
where $N(t_{0})$ is the initial number of helium atoms in the closed pores.

Further, by dividing the section of the helium release curve after the shutdown of the reactor into time intervals Δ*t*, the amount of helium in the pores for each moment of time was calculated in a recursive manner:$$t_{i}=i\cdot ∆t$$
(6)$$N\left(t_{i}\right)=N\left(t_{0}\right)-\int_{t0}^{t_{i}}F\left(t\right).$$

$N\left(t_{0}\right)$ can be determined from the graph in Figure 3 by integrating the entire helium release curve; the value of $\int_{t0}^{t_{i}}F(t)$ can be determined by integrating the helium release curve from the initial moment to time $t_{i}$.

From the obtained values it is possible to determine the parameter *β*($t_{i}$) for each time $t_{i}.$
(7)$$\beta \left(t_{i}\right)=\frac{-\ln\left(\frac{N(t_{i+1})}{N(t_{i})}\right)}{∆t}$$

The values of *β* (obtained as a result of the calculation for the section of the helium release curve where the flux drops 10 times from the maximum value) ranged from 0.0035 to 0.005 (Figure 10). The relatively small difference in the values of *β* obtained for different sections of the helium release curve indicates a possible description of the helium release from pebbles when the internal porosity was opened by the effusion mechanism. A certain trend towards a decrease in the probability of a helium atom leaving the internal pores with a decrease in temperature was possibly associated with a decrease in the flow cross-section of effusion during thermal compression.

## 4. Conclusions

As a result of the analysis of experiments regarding the irradiation of lithium metatitanate in vacuum conditions, it has been established that during irradiation, peak releases of helium from closed pores of the ceramics have been observed, which open during the first 7 days of irradiation.

Based on the data obtained, it can be formally assumed that the result of such a mechanism is the following pattern of pore opening: at the initial stage of irradiation, larger pores are opened (with a characteristic size up to 10 μm, and more); then, for some time (up to 3–4 days of irradiation), pores with a characteristic size up to 2 μm are mainly opened; next, smaller pores (up to 1 μm) are opened; after that, on days 7–8, the processes of “free paths” formation virtually cease. However, real cracking should appear first when stresses are concentrated on the strongly curved surfaces of closed pores, and also when there is a high helium pressure within the pores.

The causes of sample cracking are temperature gradients across the ceramic sample resulting from internal heating of the pebbles under vacuum pumping conditions, and contact with the bottom of the pumped capsule of the ampoule device.

Analysis of the experiment sections associated with reactor stoppage allowed us to simulate the diffusion release of tritium from ceramic pebbles. According to the results of the simulation, the temperature dependence of the effective diffusion coefficient of tritium in ceramics at the end of irradiation was determined. Results of calculating the probability values (*β*) of the helium atom to leave the internal pores simulation of helium effusion from the pores of lithium ceramics during reactor stoppage were presented.

The values of the effective tritium diffusion coefficient reasonably coincided with the values obtained in the analysis of the experimental plots when tritium was released as a result of heating samples during their irradiation.

These newly obtained experimental data can be used to analyze the processes of tritium generation and release under different modes of operation of breeder blankets using lithium metatitanate pebbles.

## Figures and Tables

**Figure 1 materials-16-05903-f001:**
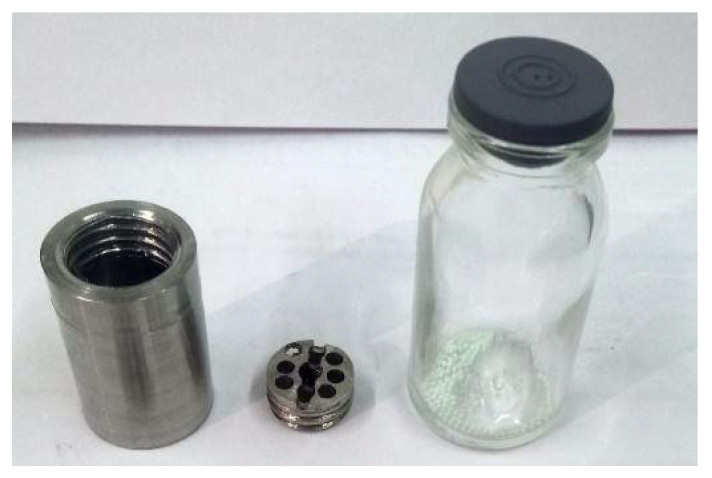
Loading capsule of the ampoule device and investigated samples.

**Figure 2 materials-16-05903-f002:**
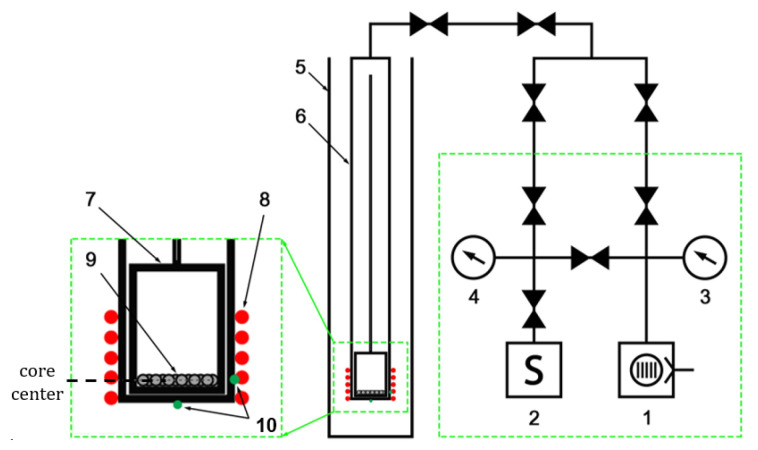
CIRRA experimental facility (loading capsule enlarged): 1—turbomolecular pump; 2—mass spectrometer; 3, 4—pressure sensors; 5—reactor experimental channel; 6—ampoule device; 7—loading capsule; 8—heater; 9—samples; 10—thermocouples.

**Figure 3 materials-16-05903-f003:**
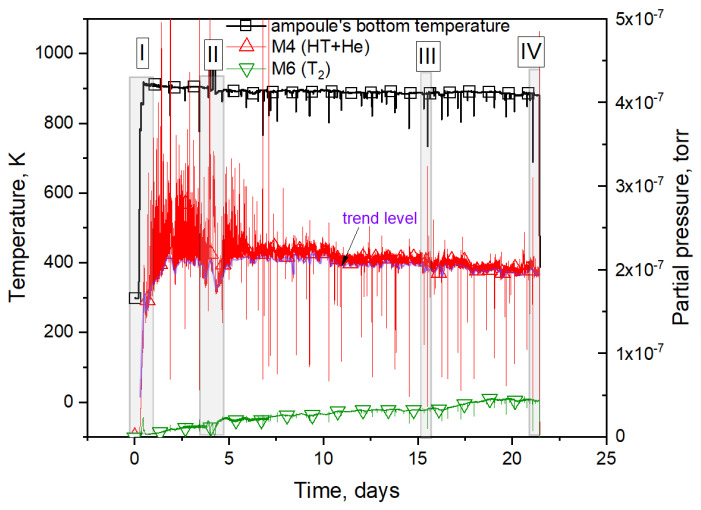
Diagram of the reactor experiment regarding the irradiation of lithium ceramics.

**Figure 4 materials-16-05903-f004:**
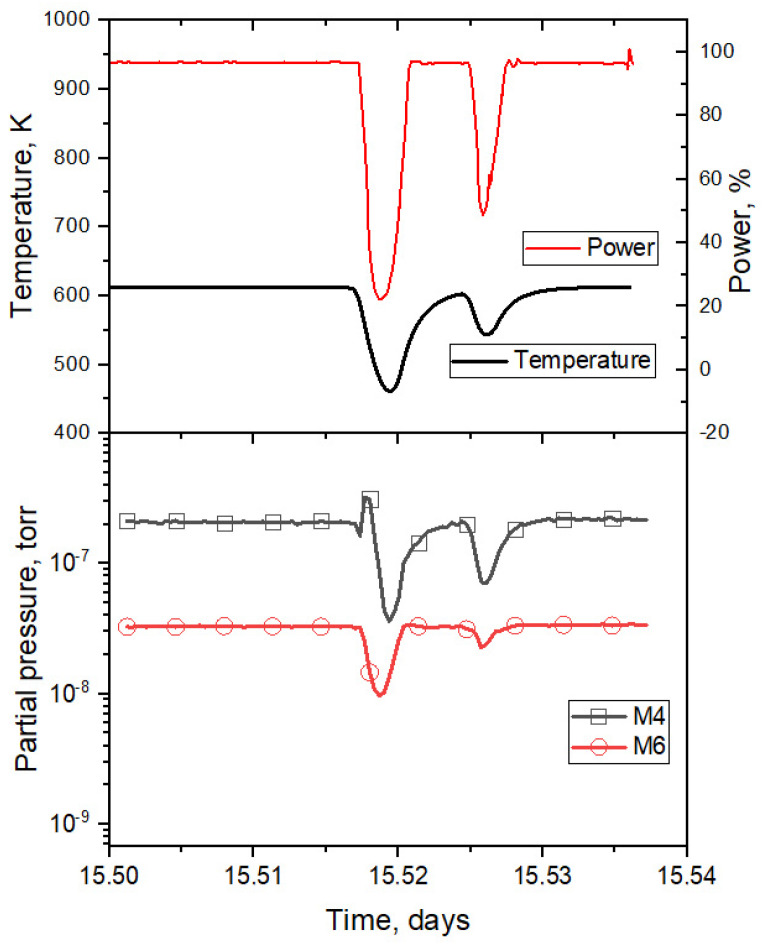
Enlarged area of one of the sections (section III) of the reactor experiment with a short-term reactor power drop [6].

**Figure 5 materials-16-05903-f005:**
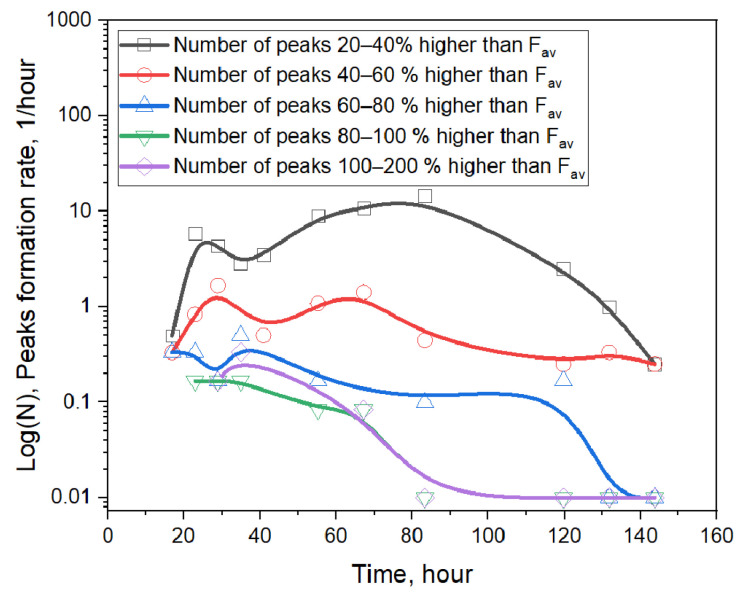
Time dependence of the rate of helium peak release from lithium ceramics.

**Figure 6 materials-16-05903-f006:**
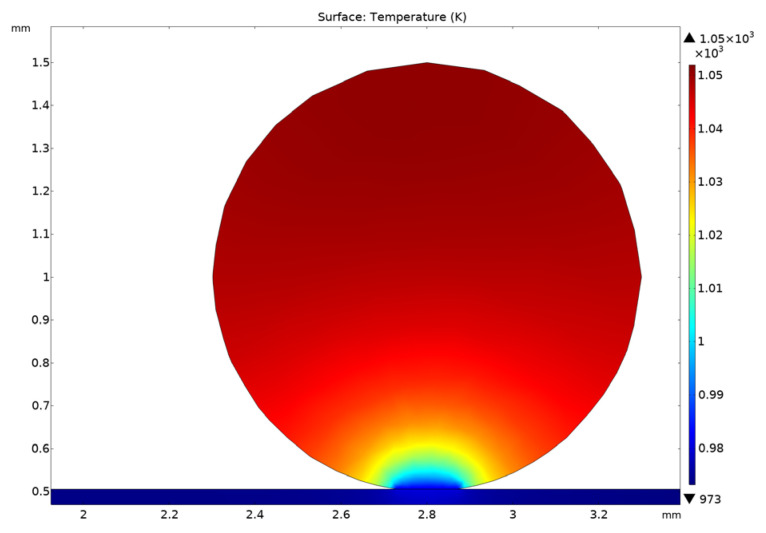
Equilibrium temperature field along the pebble of lithium metatitanate at the bottom of the capsule during irradiation.

**Figure 7 materials-16-05903-f007:**
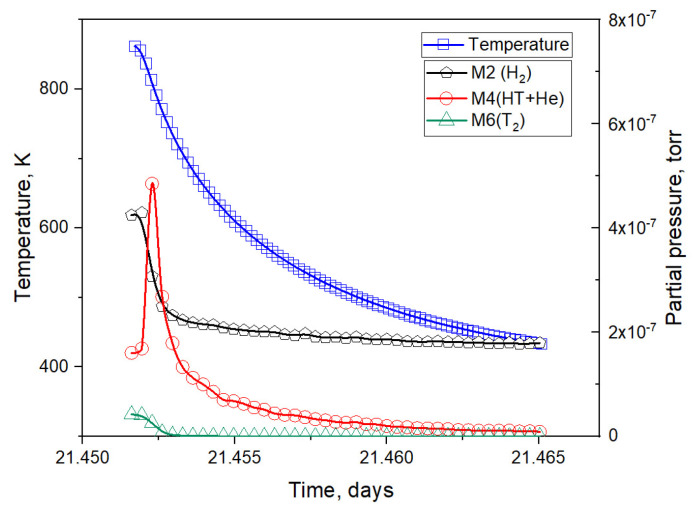
Enlarged area of the experiment section during a reactor stoppage.

**Figure 8 materials-16-05903-f008:**
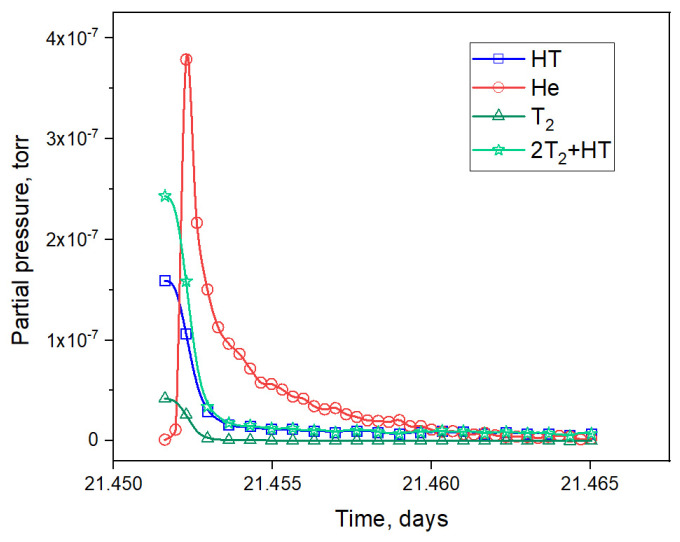
Release of various gases into the experimental chamber after a reactor stoppage.

**Figure 9 materials-16-05903-f009:**
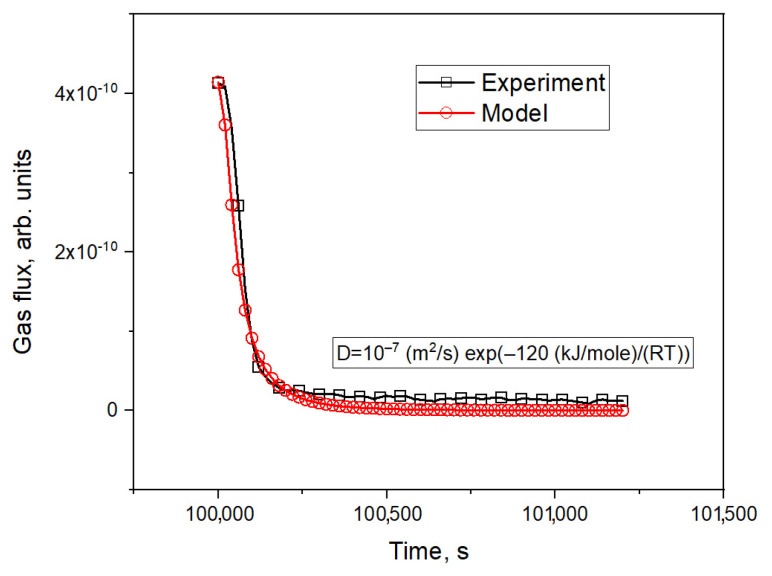
Simulation results of tritium release from lithium ceramics during reactor stoppage.

**Figure 10 materials-16-05903-f010:**
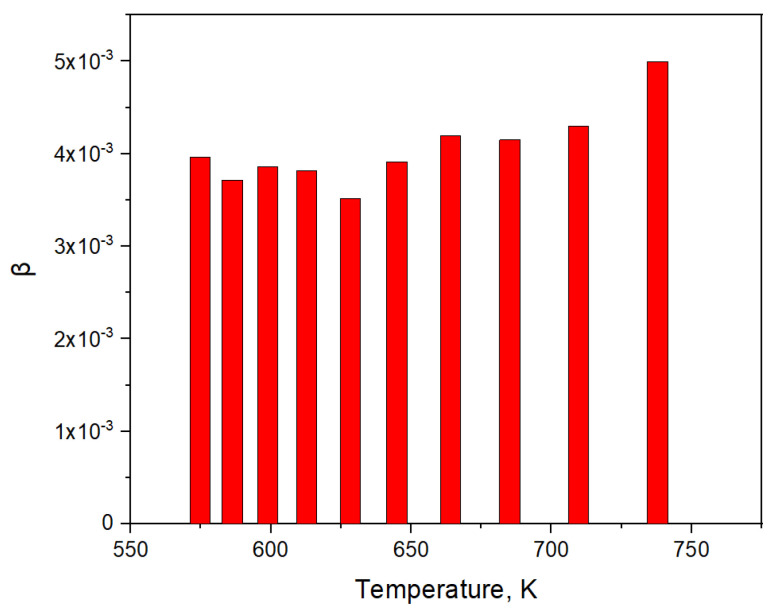
Results of calculation of the probability values (*β*) of the helium atom to leave the internal pores for different pebble temperatures during reactor stoppage.

**Table 1 materials-16-05903-t001:** Elemental composition of lithium ceramics Li_2_TiO_3_.

Elements	Li	Ti	O	Ca	Na	K	Mg	B	Co	Al	Zr	Fe
composition (w%)	11.4	44.4	44.1	<0.01	0.012	<0.0001	0.0006	<0.01	0.0030	0.009	0.0003	0.007

## Data Availability

Not applicable.

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
