# Peer review of "Features of Helium and Tritium Release from Li2TiO3 Ceramic Pebbles under Neutron Irradiation"

_materials, 2023, doi:10.3390/ma16175903_

Round 1

Reviewer 1 Report

The article presents information about helium and tritium release from Li2TiO3 ceramic pebbles occurring under fluctuations of the neutron flux. The article is well-written and thoughtful. The authors describe everything in detail also referring to previous studies. In my opinion, the authors present data and a discussion that provides concrete insights into the topic of research. The conclusions are adequate.

Authors provide in Table 1 the elemental composition of Li2TiO3 ceramics, but there is no information about the method of ceramics production – please provide some information about ceramics production.

Author Response

 The Authors are grateful to the Reviewer for helpful recommendations

Added to the manuscript:

Pebbles were obtained by the sol-gel technology.

Reviewer 2 Report

This work is a continuation of a series of works by the authors on the study of the generation and release of tritium and helium from lithium ceramics of various compositions. The manuscript is well prepared, but requires the reader to be constantly distracted by the previous work of the authors.

The submitted article is in the scope of the "Materials" journal and can be published after the elimination of several comments.

1) It would be useful to indicate the method of manufacture of the ceramics sample.

2) It is necessary to make references to the source describing the experimental facility and the WWR-K reactor, since this information is necessary.

3) Indicate in the manuscript what the designation 2*T2+HT in Figure 8 means.

4) Were studies of the microstructure of the irradiated samples of lithium metatitanate carried out? If not, is it planned to conduct them and confirm with their help the results indicated in this manuscript?

5) Since the ceramic is highly enriched (96% in lithium 6), it would be useful if the authors gave some estimates of the uniformity of pebble irradiation, did the authors take into account the effect of self-shielding of the pebble bad in the model?

6) Why graph in Figure 5 does not show data on the peak release of helium for the initial stage of irradiation (the first day of irradiation)? Does this mean that there was no peak helium release at the initial stage?

Pay attention to errors in the text:

1 line 1 line 27 6Li→6Li

2 line 3 line 105 “he” → “the”

3 line 3 line 74 Li2TiO3 → Li2TiO3

4 page 7 line 219 Polanyi-Wigner's law

7) In the discussion section, before conclusion, it will be important to give some general comparison of the observed radiation effects with that well established in other binary oxides ( MgO, alumina, spinel)

Author Response

Q1

This work is a continuation of a series of works by the authors on the study of the generation and release of tritium and helium from lithium ceramics of various compositions. The manuscript is well prepared, but requires the reader to be constantly distracted by the previous work of the authors.

The submitted article is in the scope of the "Materials" journal and can be published after the elimination of several comments.

1) It would be useful to indicate the method of manufacture of the ceramics sample.

A1

Added to the manuscript:

Pebbles were obtained by the sol-gel technology.

Q2

2) It is necessary to make references to the source describing the experimental facility and the WWR-K reactor, since this information is necessary.

A2

The reference [Asset Shaimerdenov, Shamil Gizatulin, Daulet Dyussambayev, Saulet Askerbekov &Inesh Kenzhina. The WWR-K Reactor Experimental Base for Studies of the Tritium Release from Materials Under Irradiation // Fusion Science and Technology, Volume 76, 2020 - Issue 3: Part I Pages 304-313 Published online: 07 Feb 2020 https://doi.org/10.1080/15361055.2020.1711852] has been added.

Q3

3) Indicate in the manuscript what the designation 2*T2+HT in Figure 8 means.

A3

Corrected in manuscript:

From these expressions, we can separate the release of gases with a mass number of M4 into helium and the HT molecule. In Fig. 8 is represented the release of various gases into the experimental chamber, including total flux of tritium (2T2+HT).

Q4

4) Were studies of the microstructure of the irradiated samples of lithium metatitanate carried out? If not, is it planned to conduct them and confirm with their help the results indicated in this manuscript?

A4

Unfortunately, we have not yet been able to confirm the proposed helium release mechanism from the data of microstructural studies due to the high activity of the test samples. But the microstructural studies of irradiated samples are planned.

Q5

5) Since the ceramic is highly enriched (96% in lithium 6), it would be useful if the authors gave some estimates of the uniformity of pebble irradiation, did the authors take into account the effect of self-shielding of the pebble bad in the model?

A5

Inserted in article:

Ceramic pebbles were poured into the bottom of the experimental ampoule device in 1 layer. The calculations performed showed that the self-shielding effect in the backfill is less than 10%.

Q6

6) Why graph in Figure 5 does not show data on the peak release of helium for the initial stage of irradiation (the first day of irradiation)? Does this mean that there was no peak helium release at the initial stage?

A6

At the initial stage of the experiments (stage 1, Fig. 3), the power of the reactor was increased, and, accordingly, the temperature of the samples. This section was studied in detail in [4]. We also note that during the first day of irradiation, the peak release of helium was insignificant.

Q7

Pay attention to errors in the text:

1 line 1 line 27 6Li→6Li

2 line 3 line 105 “he” → “the”

3 line 3 line 74 Li2TiO3 → Li2TiO3

4 page 7 line 219 Polanyi-Wigner's law

A7

Corrected

Q8

7) In the discussion section, before conclusion, it will be important to give some general comparison of the observed radiation effects with that well established in other binary oxides (MgO, alumina, spinel)

A8

Inserted into the manuscript:

Neutrons interacting with ceramic oxides can cause induced activation, formation of gas bubbles, reduction in thermal conductivity, embrittlement and other material degrading effects [Moriyama, H., Tanaka, S., Noda, K. Irradiation effects in ceramic breeder materials, J. Nucl. Mater. 1998, 258–263, 587-594.; .Kotomin, E.A., Popov, A.I. Radiation-induced point defects in simple oxides. Nucl. Instrum. Methods Phys. Res. 1998, 141, 1–15]. We suppose, these factors can lead to cracking of lithium metatitanate pebbles located in the contact area.

Reviewer 3 Report

Dear authors,

a generally well structured and elaborated paper on experimental and simulational investigations into gas release properties of breeder ceramics under neutron irradiation. I have a couple of comments to be addressed in a revision of the paper.

General comments:

- Line 30: A statement like "authors believe" is not appropriate for a scientific publication. I have the impression, that indeed, some of your findings might be not really established in current or referenced work. In this case, you could refer to something like assumptions which need to be confirmed in future/other work. Please remove "believe" and change it to "assume" in the abstract.

- In many figures you are showing line graphs with supposedly experimental data points (as an example Fig. 4). However, it is not clear, whether the line graphs actually are continuous measurements, which I assume e.g. in Fig. 4. Even more confusing is this in Fig 5. How does here the dots link to the lines, are lines some kind of fits or spline approximations?

- In section 3.1 you are referring to peak/drops of release relative to a trend level. Please discuss and plot this trend level curve. You are also refering to an average release level (or gas pressure level) in section 3.2, which should be better linked to the 3.1.

- It would be instructive also to discuss a release peak (integral release in a peak) vs. trend/average, but I guess this is done in ref. 8,9. Could you confirm.

- Line 175 following: I think you eman the helium release "in a single peak" is proportional to surface and irradiation time. It is not clear, how you estimate the average diameter (not volume) of the pores in the material. Do you assume all generated He has been released by the end of irradiation?

- Could you explain the rationale of opening large pores first? I would expect the cracking should appear primarly at stress concentrations at highly curved surfaces possibly with a high He pressure.

Specific comments:

- Line 70: please introduce the mass of 4 acronym.

- Line 105: typo "he" should the "the".

- Figures: I would strongly suggest to use line graphs only, when individual data points are dense (e.g. Fig. 3

- Fig. 5: black/red lines legend are the same, should be diffent. Rather than "exceeding" it is meant "falling into" (an intervall).

- line 191: your reference to 10,11 does not give any information regarding your statement on pebble cracking at contact surfaces. Please check.

- Line 200: the gradient across a pebble seems to be less than the stated 100 K/mm. However, could you explain how this gradient is relevant, should it not be the peak gradient close to the contact surface.

- Line 217: use superscript 6Li.

Good language style, no specific comments; see some tiny improvement suggestion in the other comments.

Reviewer 4 Report

The manuscript “Features of helium and tritium release from Li2TiO3 ceramic pebbles under fluctuations of the neutron fluxis” devoted to the study of the processes of helium outflow and tritium diffusion in breeder blankets with lithium metatitanate pebbles irradiated with fast neutrons. These studies continue and develop the cycle of their numerous publications of the authors of this manuscript on the generation of tritium during the operation of a D-T fusion reactor.

In this manuscript, the authors came to the conclusion about a stepwise opening of pores in which implanted helium accumulates over seven days, and they conclude that a opening time of the pores is determined by their sizes from 1 to 10 μm. Unfortunately, this statement is confirmed only by general reasoning and rather conditional and not always convincing assessments. It should be noted that in one of their previous papers [(Kulsartov, T., Tazhibayeva, I., Gordienko, Y., Chikhray, E., Tsuchiya, K., Kawamura, H., & Kulsartova, A. (2011). Study of tritium and helium release from irradiated lithium ceramics Li2TiO3. Fusion Science and Technology, 60(3), 1139-1142.] they declare the diffusion nature of helium release of from the volume of lithium metatitanate pebbles. There is no reference to this work in this manuscript. In my opinion, it is necessary either to refute the results of 2011, or to consider both mechanisms of helium outflow.

The explanation of the "free pathways formation" process by cracking and stepwise "irradiation pores" of various sizes requires direct confirmation. In particular, it is desirable to present the micrographs of the cross section of the pebbles under study as it done, for example, in [Hong, M., Zhang, Y. C., Liu, Y. H., & Fu, B. J. (2012). Fabrication of Li2TiO3 ceramic pebbles by gelcasting method. Key Engineering Materials, 512, 1717-1720.]

There are also comments on the drawings. For example, in Figures 7 and 8, the graphs of dependencies in the maxima are based on three experimental points, which is clearly not enough to describe these dependencies. Moreover, the first and second points in these dependencies are connected not by a straight line, but by a broken line with a sharp bend almost at a right angle. The simulation presented in Figure 10 does not correspond to the experimental data either in terms of the form of the dependence or in terms of numerical values. In some ranges, the experimental and model data differ by a factor of two or more.

It is also necessary to correct the title of the manuscript, which refers to "fluctuations of the neutron flux", but the text does not discuss fluctuations of the neutron flux.

This manuscript is in need of significant revision.

 Author Response

Round 2

Reviewer 2 Report

The authors have successfully improved the original version of their manuscript, responding constructively to all the comments/recommendations of the reviewer.  Therefore, the article can be recommended for publication.

Author Response

On behalf of all authors, we express our gratitude for reviewing the article

Reviewer 4 Report

I thank the authors for their work on improving the text of the manuscript. I understand their difficulties with measurements and characterization of samples.

The article may be approved for publication after the following corrections have been made.

1. In Figures 7 and 8 there should be no broken curves connecting adjacent experimental points. The figure captions should indicate that the solid lines on the dependencies are only guidelines for the eyes.

2. The authors should admit that the model they proposed does not adequately describe the experimental dependences. In most of the time range, the difference between the experiment and the model in Figures 9 and 10 reaches several times. The starting point of the time range in Figures 9 and 10 is different, but this is not explained in the text. It is possible that a model with other parameters will be able to explain the initial parts of the experimental dependences. A substantiation of the significant difference between the model and experiment in the range from 200 seconds and further should be provided. An attempt to explain the differences between the experiment and the model due to the scatter of the measurement results is incorrect, the measured dependences have a monotonous form. Negative values on the y-axis have no physical meaning and should not be present in the figure.

Author Response

Please see the attachment."
